# Ivermectin Does Not Protect against SARS-CoV-2 Infection in the Syrian Hamster Model

**DOI:** 10.3390/microorganisms10030633

**Published:** 2022-03-16

**Authors:** Caroline S. Foo, Rana Abdelnabi, Laura Vangeel, Steven De Jonghe, Dirk Jochmans, Birgit Weynand, Johan Neyts

**Affiliations:** 1KU Leuven Department of Microbiology, Immunology and Transplantation, Rega Institute for Medical Research, Laboratory of Virology and Chemotherapy, B-3000 Leuven, Belgium; caroline.foo@kuleuven.be (C.S.F.); rana.abdelnabi@kuleuven.be (R.A.); laura.vangeel@kuleuven.be (L.V.); steven.dejonghe@kuleuven.be (S.D.J.); dirk.jochmans@kuleuven.be (D.J.); 2Global Virus Network, GVN, 725 West Lombard St, Room S413, Baltimore, MD 21201, USA; 3KU Leuven Department of Imaging and Pathology, Translational Cell and Tissue Research, Division of Translational Cell and Tissue Research, B-3000 Leuven, Belgium; birgit.weynand@uzleuven.be

**Keywords:** COVID-19, ivermectin, molnupiravir, off-label use, hamster model

## Abstract

Ivermectin, an FDA-approved antiparasitic drug, has been reported to have in vitro activity against SARS-CoV-2. Increased off-label use of ivermectin for COVID-19 has been reported. We here assessed the effect of ivermectin in Syrian hamsters infected with the SARS-CoV-2 Beta (B.1.351) variant. Infected animals received a clinically relevant dose of ivermectin (0.4 mg/kg subcutaneously dosed) once daily for four consecutive days after which the effect was quantified. Ivermectin monotherapy did not reduce lung viral load and even significantly worsened SARS-CoV-2-induced lung pathology. Additionally, it did not potentiate the activity of molnupiravir (Lagevrio^TM^) when combined with this drug. This study contributes to the growing body of evidence that ivermectin does not result in a beneficial effect in the treatment of COVID-19. These findings are important given the increasing, dangerous off-label use of ivermectin for the treatment of COVID-19.

## 1. Introduction

Two years after it was declared a global pandemic by the World Health Organization, the coronavirus disease (COVID-19) continues its devastating impact, having claimed officially over 6 million lives as of 16 March 2022 [1]. The etiological agent, the severe acute respiratory syndrome coronavirus 2 (SARS-CoV-2), first emerged in Wuhan, China in December 2019 and has since evolved into variants emerging from different regions of the world, namely the Alpha, Beta, Delta, and more recently Omicron variants of concern (VoC) [2,3]. These VoC have been implicated in a resurgence of infections and mortality, with increased transmissibility and potential escape from both vaccine- and natural-infection-induced immunity [3,4,5].

Ivermectin is an FDA-approved, broad-spectrum antiparasitic drug with a range of other activities, including antiviral and host immunomodulation effects [6]. This widely accessible drug has been attracting increasing off-label use for COVID-19 since its in vitro activity against SARS-CoV-2 was reported [7], which is of growing concern. In an in vivo model of mouse hepatitis virus, a type 2 family RNA coronavirus similar to SARS-CoV-2, ivermectin reduced the viral load and protected mice from disease [8]. In Syrian hamsters infected with SARS-CoV-2, treatment with ivermectin did not result in a direct antiviral activity, but rather apparently showed some immunomodulatory effects, improved lung pathology and clinical presentations [9]. Ivermectin is being assessed as a treatment for COVID-19 in a large number of clinical trials, with no clear evidence of its clinical benefit so far [10].

Molnupiravir (Lagevrio^TM^, EIDD-2801) is the orally bioavailable counterpart of the ribonucleoside analogue N4-hydroxycytidine (NHC, EIDD-1931), which was initially developed for the treatment of influenza [11]. NHC exerts a broad-spectrum antiviral activity against multiple RNA viruses of different families by its incorporation into viral RNA, resulting in the accumulation of deleterious mutations in the nascent viral RNA, and consequently, error catastrophe [12]. In pre-clinical infection models in mice, Syrian hamsters, and ferrets, molnupiravir demonstrated efficacy against SARS-CoV-2 [13,14,15], including against several VoC [16]. The drug has been approved in many regions for treatment of COVID-19.

Here, we evaluate the antiviral efficacy of ivermectin against SARS-CoV-2 Beta (B.1.351) VoC in a hamster infection model alone or in combination with molnupiravir; this combination was inspired by the observation of a pronounced combined activity of molnupiravir and favipiravir [17].

## 2. Materials and Methods

Ivermectin was purchased from Alfa Aesar (Taiwan, China) and was first formulated as 10 mg/mL solution in propylene glycol/glycerol formal (60:40 *v*/*v*), both from Sigma (Shanghai, China) then diluted to 0.1 mg/mL stock in phosphate buffered saline (PBS). Molnupiravir (EIDD-2801) was purchased from Excenen Pharmatech Co., Ltd. (Guangzhou, China) and was formulated as 50 mg/mL stock in a vehicle containing 10% PEG400 (Sigma) and 2.5% Kolliphor-EL (Sigma) in water. The formulations of both compounds were selected based on literature [13,18].

All virus-related work was conducted in the high-containment BSL3 facilities of the KU Leuven Rega Institute (3CAPS) under licenses AMV 30112018 SBB 219 2018 0892 and AMV 23102017 SBB 219 2017 0589 according to institutional guidelines. The hamster infection model of SARS-CoV-2 has been described before [19]. Briefly, 6–8 weeks female SG hamsters were treated with vehicle, ivermectin (0.4 mg/kg, once daily (QD) subcutaneously), molnupiravir (EIDD-2801, 150 mg/kg, BID orally) or combination of both drugs (ivermectin 0.4 mg/kg QD+EIDD-2801 150 mg/kg BID) for four consecutive days starting one hour before intranasal infection with 50 µL containing 1 × 10^4^ TCID_50_ of SARS-CoV-2 Beta B.1.351 variant (derived from hCoV-19/Belgium/rega-1920/2021; EPI_ISL_896474, 2021-01-11) [19]. At day four post-infection (pi), the animals were euthanized for sampling of the lungs and further analysis by i.p. injection of 500 μL Dolethal (200 mg/mL sodium pentobarbital). Lungs were collected for quantification of subgenomic viral RNA using N2 primers and probes targeting the viral nucleocapsid [20], infectious virus titers, and lung histopathology as described previously [20] (Figure 1a).

## 3. Results and Conclusions

We here tested the efficacy of single treatment with ivermectin (0.4 mg/kg, subcutaneous, once daily (QD)) or molnupiravir (150 mg/kg, oral BID) or the combination of both. Briefly, 6–8 weeks female SG hamsters were treated with the intended dose of each compound or the vehicle (i.e., the control group) for four consecutive days starting one hour before intranasal infection with SARS-CoV-2 Beta (B.1.351) variant (Figure 1a). At day 4 post-infection (pi), the animals were euthanized, and organs were collected for quantification of viral RNA, infectious virus titers, and lung histopathology (Figure 1a). Monotherapy with ivermectin (0.4 mg/kg, once daily) did not result in any significant reduction of viral RNA (Figure 1b) or infectious virus titers (Figure 1c) in the lung of treated hamsters as compared to the vehicle control. On the other hand, single treatment with molnupiravir (150 mg/kg BID) significantly reduced viral RNA and infectious virus titers in the lungs by 1.1 (*p* = 0.0048) and 1.8 (*p* = 0.0005) log_10_/mg lung tissue, respectively (Figure 1b,c). Treatment of infected hamsters with a combination of molnupiravir (150 mg/kg, BID) and ivermectin (0.4 mg/kg, once daily) resulted in a reduction of the lung viral RNA loads by 1.1 log_10_ (*p* = 0.022, Figure 1b) and lung infectious virus titers by 1.9 (*P* = 0.0002, Figure 1c) log_10_/mg lung tissue, which is similar in potency to the single treatment with molnupiravir (150 mg/kg, BID). No significant weight loss or toxicity signs were observed in any of the treated groups (Figure 1d). However, the average %body weight change on the day of the sacrifice (compared to day zero) in the ivermectin single treatment group (average of 1.7%) was markedly lower than the vehicle-treated group (average of 2.6%), Figure 1d.

The median histopathology score in the single ivermectin treatment group (5.8) was significantly higher than the vehicle treated control (median score of 4.5, *p* = 0.04) (Figure 2a). A significant improvement in lung pathology was observed in the single molnupiravir-treated group (median score of 3.5, *p* = 0.04), while the combined molnupiravir/ivermectin treatment resulted in a median histopathology score of 3.8 (Figure 2a). Hematoxylin/eosin (H&E)-stained images of lungs of the ivermectin-treated hamsters revealed severe perivascular inflammation with vasculitis and significant multifocal bronchopneumonia (Figure 2b). On the other hand, the lungs of animals treated with molnupiravir as a single treatment showed very limited peribronchial and perivascular inflammation (Figure 2b).

In our study, ivermectin did not result in any antiviral activity against SARS-CoV-2 in Syrian hamsters as monotherapy. This is consistent with a previous study published also in hamsters, wherein ivermectin was dosed once subcutaneously at 0.4 mg/kg (an antiparasitic dose used in the clinical setting) and monitored for four days [9]. However, unlike the previous study, we did not notice an improvement in SARS-CoV-2-induced lung pathology with ivermectin. Transcriptomic profiling of the lungs was performed in the previous study to compare expression levels of inflammatory genes with vehicle and ivermectin treatment, as well as a descriptive histopathological analysis which indicated that SARS-CoV-2-infected, ivermectin-treated hamsters exhibited reduced degrees of edema and congestion, with greater numbers of mononuclear cells in the alveolar spaces [9]. In our study, a cumulative lung score was performed based on ten different parameters, including edema and congestion, which had similar scorings in both vehicle-treated and ivermectin-treated groups. Additionally, no mononuclear cells in the alveolar spaces were observed. These differences in findings may be due to slight hamster age differences or virus strain differences used for infection, namely that the SARS-CoV-2 Beta variant was used in our study and the ancestral SARS-CoV-2 strain (BetaCoV/France/IDF00372/2020) was used in the previous study [9].

Additionally, ivermectin did not potentiate the activity of molnupiravir in our study, indicating that it is unlikely to have a significant role in directly or indirectly modulating SARS-CoV-2 replication. A C_max_ of 80.2 ng/mL was measured in Syrian hamsters when treated once subcutaneously with 0.4 mg/kg of ivermectin [21], comparable to that measured in humans when dosed at similar levels orally [9,22]. Taken together with other available evidence, it suggests that in order to achieve therapeutic activity against SARS-CoV-2 (IC_50_ in Vero cells of 1750 ng/mL [7]), the dosing levels of ivermectin in humans would need to be substantially increased from the approved clinical dose to potentially toxic doses [23,24,25]. Recently, it has also been reported that ivermectin can inhibit SARS-CoV-2 in Vero cells, but not in human-airway-derived cells [26]. This is in line with the fact that several studies claiming the clinical benefits of ivermectin in COVID-19 have been found to be flawed [27], and several better-designed trials such as IVERCOR-COVID19 (NCT04529525) [28], I-TECH (NCT04920942) [29] and NCT04405843 [30] have demonstrated that ivermectin does not have clinical efficacy against COVID-19 and does not significantly prevent progression to severe disease or hospitalization.

In conclusion, we report that ivermectin, at a clinically relevant dose, does not reduce lung viral load nor improve lung pathology in Syrian hamsters infected with SARS-CoV-2 Beta variant. In fact, virus-induced lung pathology increases in severity in ivermectin-treated animals. Moreover, the drug does not potentiate the antiviral activity of molnupiravir (Lagevrio^TM^). This study contributes to the growing body of evidence that ivermectin does not result in a clinical beneficial effect in the treatment of COVID-19 at the current approved dose. Hence, we strongly advise against the increasing, dangerous off-label use of ivermectin for the treatment of COVID-19.

## Figures and Tables

**Figure 1 microorganisms-10-00633-f001:**
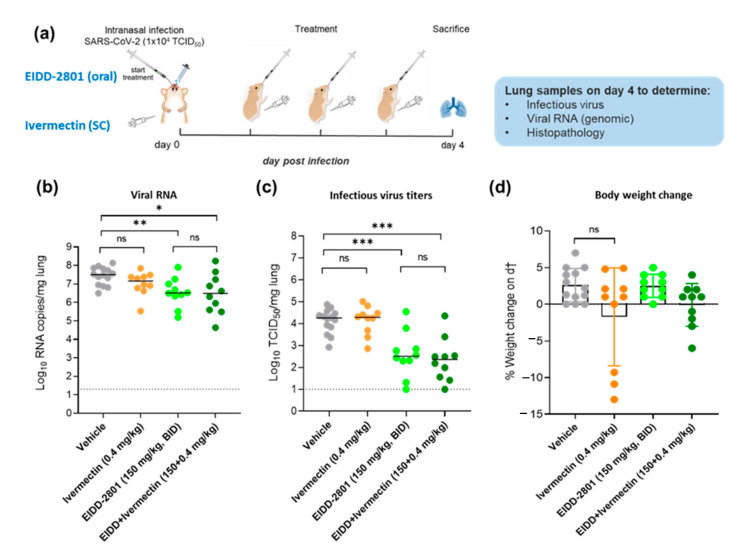
Antiviral efficacy of ivermectin as a single or combination therapy against SARS-CoV-2 beta (B.1.351) variant in a hamster infection model. (**a**) Setup of the study. (**b**) Viral RNA levels in the lungs of control (vehicle-treated), ivermectin-treated (0.4 mg/kg, QD), EIDD-2801-treated (150 mg/kg, BID), and combination-treated (EIDD-2801+ivermectin at 150 mg/kg BID+0.4 mg/kg QD) SARS-CoV-2 (B.1.351)-infected hamsters at day 4 post-infection (pi) are expressed as log10 SARS-CoV-2 RNA copies per milligram lung tissue. Individual data and median values are presented. (**c**) Infectious viral loads in the lungs of control (vehicle-treated), ivermectin-treated, EIDD-2801-treated and combination-treated (EIDD-2801+ivermectin) SARS-CoV-2-infected hamsters at day 4 pi are expressed as log_10_ TCID_50_ per milligram lung tissue. Individual data and median values are presented. (**d**) Weight change at day 4 pi in percentage, normalized to the body weight at the time of infection. Bars represent means ± SD. Data were analyzed with the Mann−Whitney U test. * *p* < 0.05, ** *p* < 0.01, *** *p* < 0.001, ns = non-significant. EIDD=EIDD-2801. All data (panels **b**, **c**, **d**) are from two independent experiments with *n* = 14 for vehicle group and *n* = 10 for the other groups.

**Figure 2 microorganisms-10-00633-f002:**
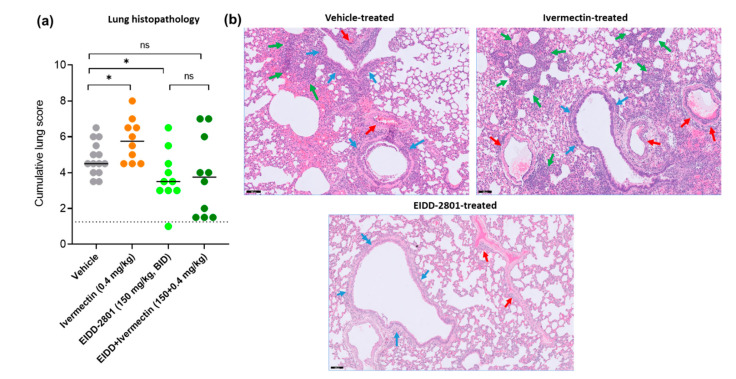
Lung histopathology of SARS-CoV-2-infected Syrian hamsters following treatment with ivermectin as a single or combination therapy. (**a**) Cumulative severity score from H&E stained slides of lungs from control (vehicle-treated ivermectin-treated (0.4 mg/kg, QD), EIDD-2801-treated (150 mg/kg, BID) and combination-treated (EIDD-2801+ivermectin at 150 mg/kg BID+0.4 mg/kg QD) SARS-CoV-2 (B.1.351)-infected hamsters at day 4 post-infection (pi). Individual data and median values are presented, and the dotted line represents the median score of untreated non-infected hamsters. Data were analyzed with the Mann−Whitney U test. * *p* < 0.05, ns = non-significant. Data are from two independent experiments with *n* = 14 for vehicle group and *n* = 10 for the other groups. EIDD = EIDD-2801. (**b**) Representative H&E images of lungs of control (vehicle-treated), ivermectin-treated and EIDD-2801-treated SARS-CoV-2 (B.1.351)-infected hamsters at day 4 post-infection (pi). Lung from vehicle control group showed peribronchial inflammation with intrabronchial cell debris (blue arrows), perivascular inflammation with endothelialitis (red arrows), and bronchopneumonia (green arrows) surrounded by intra-alveolar hemorrhage. Image of the lung from the ivermectin-treated animal showed severe perivascular inflammation with vasculitis (red arrows), limited and focal peri-bronchial inflammation (blue arrows), and significant and multifocal bronchopneumonia (green arrows). On the other hand, the lung from the molnupiravir-treated animal (EIDD-2801) revealed very limited peribronchial (blue arrows) and peri-vascular (red arrows) inflammation. Scale bar 100 µM.

## Data Availability

All of the data generated or analyzed during this study are included.

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
