# Peer review of "Ivermectin Does Not Protect against SARS-CoV-2 Infection in the Syrian Hamster Model"

_microorganisms, 2022, doi:10.3390/microorganisms10030633_

Round 1

Reviewer 1 Report

In this study, Foo et al. evaluated the antiviral efficacy of ivermection against SARS-CoV-2 in a hamster infection model alone or in combination with molnupiravir. The authors found that ivermectin at clinicall relevant doses did not reduce lung viral loads nor improve lung pathology. Moreover, the drug did not potentiate the antiviral activity of molnupiravir.

The study is of high importance and technically well performed. The findings are important due to the off-label use of ivermectin for th treatment of COVID-19. There are only some minor issues for improvement:

-Did the authors check if ivermectin had an effect in vitro in their hands.

Author Response

We thank the reviewer for his valuable comments. Indeed, we had tested Ivermectin in our own in vitro assays and we found that the compound has an EC50 = 3.64 uM and CC50 = 5.69 uM in Vero E6 cells. We had also tested the compound in A549_ACE2-TMPRSS2 cells where EC50=1.1 uM and CC50=1.8 uM were obtained. No activity was observed in Huh7 cells. As seen in these results, the compound also showed a low selectivity index in the tested cell lines with the toxic levels very close to the EC50 of the compound. Recently, it has also been reported (Kumar et al., Antimicrob. Agents Chemother. 2022, 66, e0154321) that Ivermectin can inhibit SARS-CoV-2 in Vero cells but not in human airway derived cells, indicating a cell-dependent activity of the compound.

Reviewer 2 Report

In their manuscript, Foo and colleagues report the lack of activity of ivermectin against SARS-CoV-2 infection of Syrian hamsters. Specifically, they evaluated the activity of ivermectin, Molnupiravir or a combination of both over a 4-day infection. While a reduction of vRNA and virus titres could be observed for Molnupiravir, ivermectin alone or in combination failed to significantly reduce vRNA or infectious virus production. Interestingly, an increase in lung histopathology could be observed for the ivermectin treatment group as compared to vehicle. Importantly, ivermectin did not positively influence the activity of Molnupiravir.

Overall, the manuscript is well written, results clearly reported and findings appropriately discussed. This study is an important account for the lack of efficacy of ivermectin at human-relevant dose, in an animal model.

Minor comments:

  • Figures are of poor resolution, the authors should make sure that high quality images are supplied in their final version.
  • The authors may want to explain the choice for the formulations of ivermectin and Molnupiravir, to add clarity (glycol/glycerol formal vs PEG400/Kolliphor-EL)

Author Response

We thank the reviewer for his valuable comments and for giving us the chance to improve our manuscript.

Figures are of poor resolution, the authors should make sure that high quality images are supplied in their final version.

-We will upload higher resolution Figures files with the revised version.

The authors may want to explain the choice for the formulations of ivermectin and Molnupiravir, to add clarity (glycol/glycerol formal vs PEG400/Kolliphor-EL)

-Both compounds are poorly water soluble and to formulate them we searched in literature for previously published formulations. Accordingly, the formulation of molnupiravir was based on Rosenke K et. al, Nat Commun 12, 2295 (2021) while the one for Ivermectin was selected based on Lo PK et al. Vet Res Commun 4, 251-68 (1985). We have added this information (with references) to the method section in the revised manuscript for clarity.